# Treatment of Peri-Implantitis—Electrolytic Cleaning Versus Mechanical and Electrolytic Cleaning—A Randomized Controlled Clinical Trial—Six-Month Results

**DOI:** 10.3390/jcm8111909

**Published:** 2019-11-07

**Authors:** Markus Schlee, Florian Rathe, Urs Brodbeck, Christoph Ratka, Paul Weigl, Holger Zipprich

**Affiliations:** 1Private Practice and Department of Maxillofacial Surgery, Goethe University, 60590 Frankfurt am Main, Germany; markus.schlee@32schoenezaehne.de; 2Private Practice and Department of Prosthodontics, Danube University, 3500 Krems, Austria; florian.rathe@32schoenezaehne.de; 3Private Practice, 8051 Zürich, Switzerland; ursbrodbeck@bluewin.ch; 4Department of Prosthodontics, Goethe University, 60590 Frankfurt am Main, Germany; ratka@med.uni-frankfurt.de (C.R.); weigl@em.uni-frankfurt.de (P.W.)

**Keywords:** periimplantitis, electrolytic cleaning, augmentation, air flow, re-osseointegration, classification of bone defects

## Abstract

Objectives: The present randomized clinical trial assesses the six-month outcomes following surgical regenerative therapy of periimplantitis lesions using either an electrolytic method (EC) to remove biofilms or a combination of powder spray and electrolytic method (PEC). Materials and Methods: 24 patients with 24 implants suffering from peri-implantitis with any type of bone defect were randomly treated by EC or PEC. Bone defects were augmented with a mixture of natural bone mineral and autogenous bone and left for submerged healing. The distance from implant shoulder to bone was assessed at six defined points at baseline (T0) and after six months at uncovering surgery (T1) by periodontal probe and standardized x-rays. Results: One implant had to be removed at T1 because of reinfection and other obstacles. None of the other implants showed signs of inflammation. Bone gain was 2.71 ± 1.70 mm for EC and 2.81 ± 2.15 mm for PEC. No statistically significant difference between EC and PEC was detected. Significant clinical bone fill was observed for all 24 implants. Complete regeneration of bone was achieved in 12 implants. Defect morphology impacted the amount of regeneration. Conclusion: EC needs no further mechanical cleaning by powder spray. Complete re-osseointegration in peri-implantitis cases is possible.

## 1. Introduction

Increasing numbers of inserted dental implants cause an increasing number of infected implants [1]. Mucositis is a reversible inflammatory process limited to peri-implant soft tissue. Peri-implantitis (PI) is defined as an inflammatory process affecting peri-implant hard as well as soft tissue. Typical cup-shaped progressive bone defects, pus, and bleeding on probing (BoP) are clinical parameters which have to be verified simultaneously to justify the diagnosis of PI [2,3]. Mucositis and PI are correlated with bacterial biofilms colonizing the surfaces of implants or abutments [4]. In view of the difficulty in differentiating pathologic bleeding from bleeding caused by improper probing, as well as the discordance in the dental community about the acceptable threshold of bone loss, there is no consensus about when pathology starts and how PI can be diagnosed precisely. Hence, prevalence data vary from author to author [5,6]. Based on these data, up to 100 million dental implants may be infected worldwide. As implant surfaces, which are exposed to the oral cavity, are immediately colonized by the individual microbiome, the surfaces need to be re-osseointegrated for positive long-term results [7]. Treatment and replacement of implants cause immense costs and discomfort. Therefore, it is necessary to find more effective approaches to decontaminating infected implants.

Several methods, all of which are ablative for removing biofilms have been discussed. For example, mechanical debridement by hand or ultrasound-driven curettes, brushes, lasers, pellets, cold plasma, or air-powder sprays in conjunction with or without disinfection or antibiotic agents. Re-osseointegration of between 39% and 46% of treated implant surfaces was reported in a review [8]. Re-osseointegration in humans has not yet been proven. A review of the literature demonstrated that none of the assessed methods was superior to any other in removing the biofilm and no method was able to achieve a stable result over time [9]. Up to 100% relapse of the disease for some methods was demonstrated after one year, and evidence for the superiority of any treatment modality is lacking [10]. Powder spray systems (PSS) are commonly used to treat PI. Small particles (erythritol or glycine) are accelerated by air pressure and remove the biofilm when impacting the implant surface. Several animal studies investigating re-osseointegration after cleaning by PSS proved incomplete re-osseointegration [8]. Clinical parameters like BoP, pocket depth (PD), and pus improved. Furthermore, PSS failed to prove superiority to any other treatment modality [11,12,13]. Possible reasons for this limited efficacy might be craterlike bone defects with compromised access, thus improper working angle and distance of the device, macro- and micro-design of the implant surface, and particles too large for much smaller bacteria hidden in the microstructure of textured implants. 

In an in vitro test, two bacterially contaminated implants were embedded in an electro-conductive gelatin block and were loaded with a continuous current of 0-10 mA – one acting as a cathode, the other as an anode. A reduction of bacteria was proved. This approach dramatically changed the pH at both implants [14]. Zipprich et al. covered implants with a mature biofilm, loaded them as cathode, and flooded the implants with a buffered potassium iodine solution which had passed an anode. Complete removal of the biofilm was demonstrated by SEM analysis [15]. The mode of action was investigated by a collaborative working group [16]. In an in vitro test, Ratka et al. used EC versus PSS to treat implants with different surfaces and alloys covered with a mature biofilm. In contrast to PSS, no bacteria could be cultivated in the EC groups. The difference was extremely significant [17]. Based on these findings, an electrolytic device was developed by the authors to remove the biofilm in a clinical setting. 

The aim and endpoint of this controlled clinical trial was to compare the effectiveness of two processes of decontamination in terms of bone level changes. 

## 2. Materials and Methods

### 2.1. Legal

The study was registered (BfArM DA/CA99, DIMDI 00010977) and approved by the “Ethik-Kommission der Bayerischen Landesärztekammer” (BASEC_No. DE/EKBY10) with the registration code 17075.

### 2.2. Sample Size Calculation

The data presented in this study were collected for a proof of principle study assessing the bacterial load before and after the treatment. This article describes the six-month results of the proof of principle study. Based on previous in vitro tests using a paired *t*-test with a power of 90% and a level of significance of 5%, a sample size of 12 per group was calculated. The sample size calculation was done using G*Power 3.1 (Heinrich Heine University of Düsseldorf). 

### 2.3. Devices and Mode of Action

For the electrolytic approach (EL) the implant has to be loaded negatively with a voltage and a maximum current of 600 mA. This is achieved by a device (GS1000, GalvoSurge Dental AG, Widnau, Switzerland) providing the voltage and pumping a sodium formiate solution through a spray-head, which has to be pressed into the implant by finger pressure to achieve an electrical contact (Figure 1). Driven by a peristaltic pump, a sodium formiate solution passes an anode inside the spray-head and then covers the implant with a “film” of liquid (Figure 2). The current splits the water into hydrogen anions and cations. The cations penetrate the biofilm and take an electron from the implant. Hydrogen bubbles lift the biofilm off the implant surface.

### 2.4. Patient and Sample Selection, Randomization

24 patients with at least one titanium implant and diagnosed with periimplantitis (definition according to Berglund et al.) [3] were included in the study. If more than one implant was affected, one implant was chosen randomly. The patients were allocated to test group (EC) or control group (PEC) after randomization by using sealed envelopes immediately before surgery.

### 2.5. Inclusion Criteria

Patients older than 18 years, capable of understanding and signing an informed consent, smoking fewer than 10 cigarettes per day, without uncontrolled periodontitis, BoP < 20%, Plaque Index < 20%, no allergy to the drugs or materials used, and not pregnant or nursing were suitable to be enrolled in the study. 

In contrast to most of the literature, the bone defects were not limited to intraosseous defects. All implants were included independently of their bone defect morphology, three-dimensional implant position, e.g. implant axis, inter-implant distance, etc.

### 2.6. Procedures and Measurements

Selected patients were instructed and motivated regarding proper oral hygiene and, if necessary, underwent periodontal treatment. Standardized photos were taken (occlusal, buccal, lingual view) and repeated in all appointments listed below. Suprastructures were removed 14 days before surgery. The implants were cleaned by powder spray (Nozzle, EMS, Nyon, Switzerland) and rinsed with chlorhexidine (Chlorhexamed forte, GlaxoSmithKline, Munich, Germany) to reduce soft tissue inflammation in line with standard procedures in periodontal therapy. A cover screw was placed. As a result of this pretreatment, in most cases the soft tissue grew over the implant, leaving a crestal fistula. PD and BoP were assessed at six defined points (m, mb, b, db, d, dl) (Figure 3) using a periodontal probe with a 1 mm scale (PCPUNC 15, HuFridy, Chicago, IL, USA). A crestal incision with releasing incisions was performed to enable a flap to be reflected so that access to the implant could be gained. Buccal and, in the mandible, additional lingual periosteal incisions were made so that the tissues could be mobilized over the implant. The granulation tissues were removed and, if applicable, tartar or cement remnants were removed mechanically by the use of curettes (DSC13/14, HuFridy, Chicago, IL, USA) and/or ultrasonic devices (Dentsply Sirona, Bensheim, Germany). This is necessary because the EC process can only work if the electrolyte is in direct contact with the conductive implant. The distance from implant platform to the most apical position of bone (P-B) was assessed as described in (Figure 3) at the same six points as PD and BoP were measured using the described periodontal probe. In the EC group, the spray-head was pressed into the implant and the GS1000 control unit started. The current was applied 5 s after the peristaltic pump was started and the electrolytic spray was initiated. The cleaning process took 120 s. In the PEC group, the implants were treated according to the manufacturer’s manual by a powder spray (Airflow Plus powder, Airflow, EMS, Nyon, Switzerland) for 60 s followed by the treatment described for the EC group. After cleaning, the implants were rinsed with sterile saline and augmented with a mixture of autogenous bone harvested from the ramus area (Micross Safescraper, Zantomed, Duisburg, Germany) and Bio-Oss (Geistlich, Wohlhusen, Switzerland) in a 50:50 ratio. In cases with non-supporting infrabony defects, tenting screws were used for space maintaining (Umbrella-Screw, Ustomed, Tuttlingen, Germany). After placement of a collagen membrane (Bio-Gide, Geistlich, Wohlhusen, Switzerland) the flap was coronally advanced to cover the site passively. The 6-0 propylene monofilaments (Medipac, Kilis, Greece) were removed after two weeks, and wound healing was documented. A VAS (pain, acceptance) assessment was also done by the patients. When no exposure was present at the time of suture removal the patients were checked again four weeks later, then scheduled six months after surgery for second-stage surgery. In the case of exposure, the patients were instructed to brush the area carefully and rinse the site with chlorhexidine. These sites were checked monthly.

During the period of healing (six months) the patients were supervised, and exposures or infections were documented. After six months, second-stage surgery was performed. The implants and the surrounding bone were exposed and P-B was assessed under direct vision at the previous points. In cases with exposures, no second-stage surgery was necessary and P-B was assessed by bone sounding under local anesthesia using the described periodontal probe with sufficient pressure. Furthermore infections, BoP, and recessions were documented. For all implants, suprastructures were replaced, photographs were taken, and a standardized x-ray in the right-angle position was performed. Sutures were removed after 14 days, if applicable. Bone levels at baseline and after six months were assessed by two examiners using software (DBS Win, Dürr, Bietigheim-Bissingen, Germany). The examiners, not knowing the aim of the study, were calibrated until their results correlated adequately as measured by Cohen’s Kappa (κ ≥ 0.6). In addition, the measurements had to reach 90% agreement for ±0.5 mm as well as exact agreement in 75% of the radiologic measurements before assessment of the x-rays in this study.

Schwarz et al. [18] introduced a classification describing typical peri-implant bone defect anatomy. For clinical purposes, however, a classification providing information about the healing potential of a defect would be more helpful for the practitioner. Therefore, we hereby introduce the RP Classification differentiating the regenerative potential (RP) of a bone defect based on the risk-chance ratio of treatment. We assessed intrabony defects (RP1), intrabony defects with dehiscence defects (RP2) and horizontal bone defects (RP3) (Figure 4) and correlated them to total and median bone fill. Complete regain of bone is dependent on the type of implant. In most of the implants 1 mm remodeling occurs within the first year. It cannot be expected that these implants will re-osseointegrate up to the platform. Implants with a polished neck osseointegrate to the border rough-polished. They are placed at this level very often. More coronal re-osseointegration cannot be expected [19,20]. 

Therefore, we define bone-to-implant contact with a P-B < 1 mm as complete bone fill. Implants with polished necks are counted as complete bone fill, if the bone fill reaches the border rough-polished being visible at T2 after flap removal. 

All the surgical procedures and clinical assessments such as PPD, recessions and BoP were performed by the first-named author.

### 2.7. Statistics

Quantitative values are presented as mean and standard deviation, and minimum and maximum, as well as quartiles. Gaussian distribution of data was assessed using the Saphiro-Wilk test. Comparisons were performed with the Mann-Whitney U test or *t*-test, as appropriate. The homogeneity of variance was verified by Levene’s test before the *t*-tests. The level of significance was set at α = 0.05 and all tests were two-sided. The statistical analysis was performed using R 3.6.1 with the package car 3.0-3. R. The package used is available from CRAN at http://CRAN.R-project.org/.

## 3. Results

The distribution of gender and age was homogeneous (12f/12m; mean age 57.13 y) and there were four smokers (3 EC, 1 PEC) < 10 cigarettes per day. All the sites were infected, BoP was positive, pus drained from pockets, and all sites probed deeper than 5 mm at baseline (Table 1). PD was 6.64 mm in the EC and 7.02 mm in the PEC group. No significant difference was assessable (Fisher’s exact test, *p* > 0.999). Thus, the entity was considered to be homogeneous. 

Nineteen sites were exposed at suture removal, 15 after six months. Nevertheless, no implant was lost during the healing phase. One implant from the PEC group had to be removed after six months because of infection, incomplete bone regeneration, and the presence of infection. Compliance of this female patient was poor, and the implant was placed far too lingually in a compromised axis. The addition of these factors led to the recommendation to remove this implant.

The quality of regained bone was assessed by visual inspection with a microscope (24x magnification), bone sounding with high pressure, and interpretation of x-rays. Complete and solid bone in direct contact with the implant was confirmed.

Bone gain was assessed at six months as a difference between P-B_baseline_ and P-B_6 months_ (ΔP-B) at the six defined points. There was no statistical significant difference either between the specific points or the median. Bone gain and differences between the EC and PEC groups are visualized in Table 2. Bone gain was 2.71 ± 1.70 mm in the EC group and 2.81 ± 2.15 for PEC. The difference (Δ PEC-EC) was not statistically significant (0.10 mm, p-value 0.87). The related boxplots are displayed in Figure 5 and Figure 6.

The implants in the study population consisted of different implant brands and designs. Their distribution is displayed in Table 3. According to our definition, complete bone fill was achieved if ΔP-B was less than 1 mm. This was observed in nine implants (37.5%). In two cases with a polished implant shoulder (Camlog, Altatec, Wimsheim, Germany), the bone fill reached the border rough-polished (ΔP-B < 2). If those cases are accepted as complete bone fill, a total of 12 implants (50%) accomplished complete bone fill. The distribution of the different defect morphologies and regenerative potential was four implants RP1, 11 implants RP2, and nine implants RP3. Complete regain of visible and probable bone up to the implant shoulder was achieved in all the RP1 implants. Six implants in RP2 cases and only two implants in RP3 achieved complete bone fill. Median bone gain related to defect morphology was 4.02 ± 0.96 mm in RP1 cases, 2.64 ± 1.58 mm in RP2 and 2.34 ± 1.58 mm in RP3 (Table 4).

In one case 1.5 mm bone grew over the top of the implant and 1.5 mm clinically hard and mature bone had to be removed to get access to the implant (Figure 7 and Figure 8) [21]. In two cases, (both RP3) the implants were covered by tartar. In one case, complete bone fill was gained. In the other case, P-B improved slightly. 

## 4. Discussion

### 4.1. Design of Study

The study was designed as a randomized and controlled trial. As none of the current treatment modalities was superior to any other and all of them failed to prove long-term stability and re-osseointegration in a clinical setting [10], they were unsuitable to serve as a control method. None of these methods was considered ethical to use. After lengthy discussion with the ethics committee, we decided to investigate whether additional ablative methods are beneficial to the outcome of the electrolytic approach. For this purpose, we cannot compare the results of this study directly with current treatment modalities. Furthermore, this RCT investigated changes in bone level clinically while most of the published articles focus on the change in clinical parameters. In view of the different endpoints, the data presented in this article cannot be compared directly with existing literature. After six months of healing after second-stage surgery, we will have the chance to assess clinical data (PPD, BoP, secretion) and compare them to existing literature. 

The definition of success in the therapy of periimplantitis is a matter of debate. Carcuac et al. approached this question by assessing three outcomes: 1. further marginal bone loss ≤ 0.5 mm, 2. outcome 1 + PD ≤ 5 mm, and 3. outcome 2 + BoP and/or suppuration on probing (SoP) = negative. According to this definition they were able to achieve 69.4%, 55.4% and 33.1% success in an RCT investigating resective surgery [22]. Assuming that elimination of a pocket should resolve the problems, these results are disheartening. In an RCT, after cleaning of the implant surface with plastic curettes and chlorhexidine gel in cases with intrabony defects, Schwarz et al. showed PD changes after four years from 7.1 mm to 4.6 mm when using Bio-Oss and a collagen membrane [23]. BoP reduced from 79 to 28%. Defect configuration seems to have a major effect on PD reduction. Schwarz et al. proved that circumferential intrabony defects exhibit a significantly higher reduction of PD compared to intrabony fenestration defects [24]. Roccuzzo et al. followed 24 patients (two dropouts) with intrabony defects in a case series for seven years. Implants were treated by mechanical debridement and application of EDTA and chlorhexidine gel followed by augmentation with Bio-Oss Collagen. The survival rate was 83.3% for SLA implants and 71.4% for TPS. PD was reduced from 6.6 ± 1.3 to 3.2 ± 0.7 mm in SLA, and 7.2 ± 1.5 to 3.4 ± 0.6 mm in TPS. BoP changed from 75.0 ± 31.2% to 7.5 ± 12.1% (SLA), and from 90.0 ± 12.9% to 30.0 ± 19.7% (TPS). The authors described successful therapy as PD ≤ 5 mm, negative BoP and no further bone loss. Success was achieved in two of 14 (14.3%) of the TPS and in seven of 12 (58.3%) of the SLA implants [25]. These results are difficult to understand as BoP, according to the definition of peri-implantitis, should have been 100% at baseline. It is not known whether non-infected implants with bone loss were enrolled in the study. This demonstrates the difficulty of using the parameter of BoP as a tool for diagnosing peri-implantitis, as discussed by Coli et al. [26]. 

### 4.2. Our Results

The potential of the electrolytic approach has previously been demonstrated in various in vitro tests [15,16,17,27]. We did not focus on intrabony defects, like the cited studies, but accepted all kinds of defects in our data. Therefore, because of the use of different endpoints, it is not possible to compare our data directly with the studies cited above. Achievement of re-osseointegration can be proved only histologically. In an animal study [27], we proved that complete re-osseointegration could be achieved after the use of EC, whereas with conventional cleaning, the bone filled the defect partially, but was never in direct contact with the implant. Clinically, bone fill cannot be equated with re-osseointegration, although this can happen with high probability after EC. The quality of the gained bone was evaluated by visual inspection, assessment of x-rays, and/or mechanical bone sounding of the most crestal position of the bone at the six predetermined points. Mature and solid bone in direct contact with the implant was detected. This still does not prove re-osseointegration. It may be stated, however, that we achieved complete bone fill (according to this definition) in all intrabony defects (RP1 cases). Complete bone fill was achieved in 50% of all cases – a number which, to the best knowledge of the authors, has not yet been quoted in the literature. In cases with exposure, no flap was raised after six months. The P-B distance could not be assessed under direct visual control. The potential bias seems to be negligible. Former studies demonstrated that even probing with 0.5 N completely lateralizes the peri-implant tissues [28]. In the study bone sounding was performed with high pressure under local anesthesia.

Published clinical studies focus on the change in radio-opalescence in x-rays. We will compare radiologic data in a further follow-up as well as PD and BoP assessments and answer the question of the long-term stability of the EC approach. 

The exposure rates (15 sites at suture removal) were much higher than is usual in the hands of the author compared to simultaneous implant placement and augmentation. This happened even though a strict initial phase was applied in order to reduce infection and inflammation. It will be necessary to investigate this issue more closely in future and possibly adapt surgical techniques. Whether the soft tissues could be compromised by the former peri-implant infection is merely a matter of speculation. 

Our in vitro data clearly prove the potential of EC for removing bacterial [17]. No data are available regarding whether periimplantitis impacts the healing potential of surrounding soft tissue. The number of exposures exceeds the number of exposures in cases of simultaneous implantation and augmentation according to the author’s experience. This requires further studies. 

Powder spray has to impact at an angle of 30–60° and a working distance of 3–5 mm according to the manufacturer’s manual. Owing to micro- and macrostructure as well as defect anatomy, this prerequisite may not always be met. In our study, we compared EC with a combination of EC and powder spray in order to assess whether additional mechanical debridement enhances outcomes. Our data support previous in vitro results, namely that EC alone is able to clean the implant surface and regain a surface onto which bone grows [17].

Tartar had to be removed in two cases before EC cleaning. It was clearly visible that the surfaces of the implants were damaged by the curettes and the ultrasonic device. Our data prove that regain of bone above the former bone level and clinical reattachment of bone are nevertheless possible. 

One implant had to be removed because of various factors (reinfection, malpositioning of the implant, and compliance of the patient). Which of these obstacles was causative for the decision is unclear. This raises the question of which implants should be treated or removed and leads to discussion about the etiology of the individual disease and the regenerative potential of this special defect. It is still a matter of discussion whether the bacterial biofilm is the only causal factor or whether bone loss caused by surgical, mechanical, or patient-related reasons, and bacterial colonization happens secondarily on the exposed surfaces [29]. This debate about etiology is not only an academic question. The success rate of possible treatments correlates with specimen susceptibility and uncorrectable surgical or mechanical obstacles. The number of implants included in the present study was too small to draw statistically significant conclusions about correlations between defect morphology and outcome. Conspicuously, all implants with completely preserved bone walls but intrabony defects (RP1) healed with complete bone fill of the defect. Only two out of nine cases with a vertical component (circumferential bone loss; RP3) achieved complete bone fill. Our data support the results of Schwarz et al. who stated that defect morphology has a major impact on healing [24]. We clearly state that the data from this study are too weak because of the sample size to justify the validity of the suggested RP classification. Initially, we did not plan to discuss the data in this way, but the results showed clear differences without reaching significance. Further studies are necessary to validate the RP classification. We treated all implants with the diagnosis of peri-implantitis. Implants placed with a bad axis, insufficient buccolingual inclination, and inter-implant distance < 3 mm were not excluded. Bone gain was smaller compared to perfectly placed implants according to the clinical impression of the authors. Statistical analysis was not performed because of small sample sizes. Further studies are recommended to clarify this issue. For treatment, planning a classification of bone defects to forecast treatment results would be helpful. Therefore, we herewith introduce a new classification focusing on risk-chance ratio: The Regenerative Potential (RP) Classification. Cup-shaped defects with complete bone walls surrounding the defect have the highest healing potential [24]. If more walls are missing and/or additional risk factors are present, removal of the infected implant may be considered. Further studies are necessary to develop a clear decision tree for determining which implants should be treated or removed and when electrolytic cleaning is helpful for long-term success. 

## 5. Conclusions

Electrolytic cleaning of contaminated implants achieves an implant surface where complete re-osseointegration is possible. This was attained in 50% of the cases. Additional mechanical cleaning by the use of powder spray devices does not improve the results further. The amount of regeneration depends on the regenerative potential of the bone (multiwall craterlike defects perform better than horizontal bone loss). Further confounding factors could not be identified owing to the limited sample size of 24 patients.

## Figures and Tables

**Figure 1 jcm-08-01909-f001:**
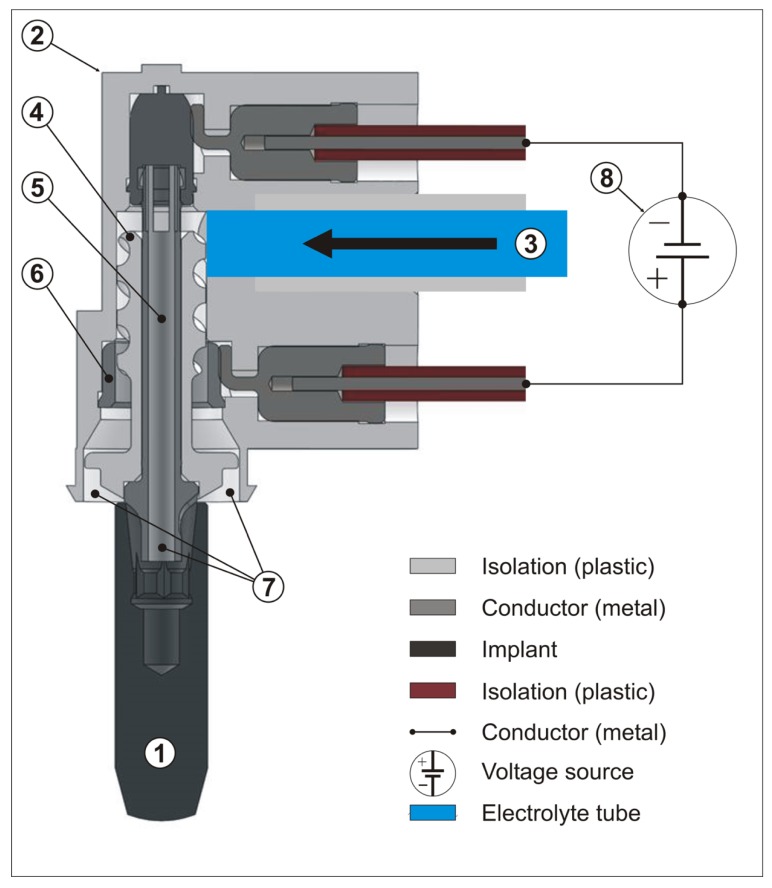
Composition of the spray-head. (1) Implant (loaded as a cathode); (2) spray head; (3) tube for electrolyte; (4) spiral-like threaded isolator; (5) connector (loaded as a cathode); (6) anode; (7) shower head (exit of electrolyte); (8) control unit and voltage source. Application of Figure 1: The spray-head (2) has to be pressed on containment of the implant (1) manually. The electrolyte will be pumped through the tube (3) and passes the spiral of the treaded isolator (4), reaches the anode (6), and will be sprayed by the shower head (7) onto the exposed implant surface. A second pathway branching off from the threaded isolator to the implant connector (5) pumps electrolyte in the implant containment (1). The positive current path derives from the voltage source (8), passes metallic conductors to the anode. The negative current path derives from the voltage source (8), passes metallic conductors to the connector (5), to the implant (1), which acts in the electrolytic process as the cathode.

**Figure 2 jcm-08-01909-f002:**
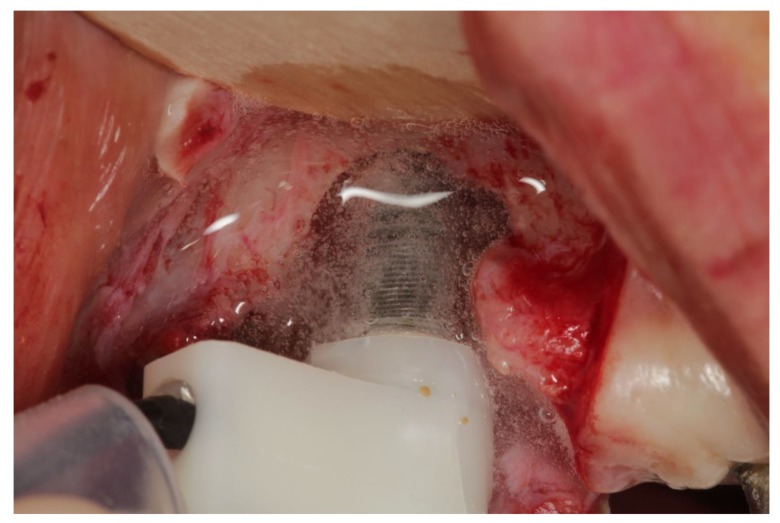
Spray-head during cleaning process.

**Figure 3 jcm-08-01909-f003:**
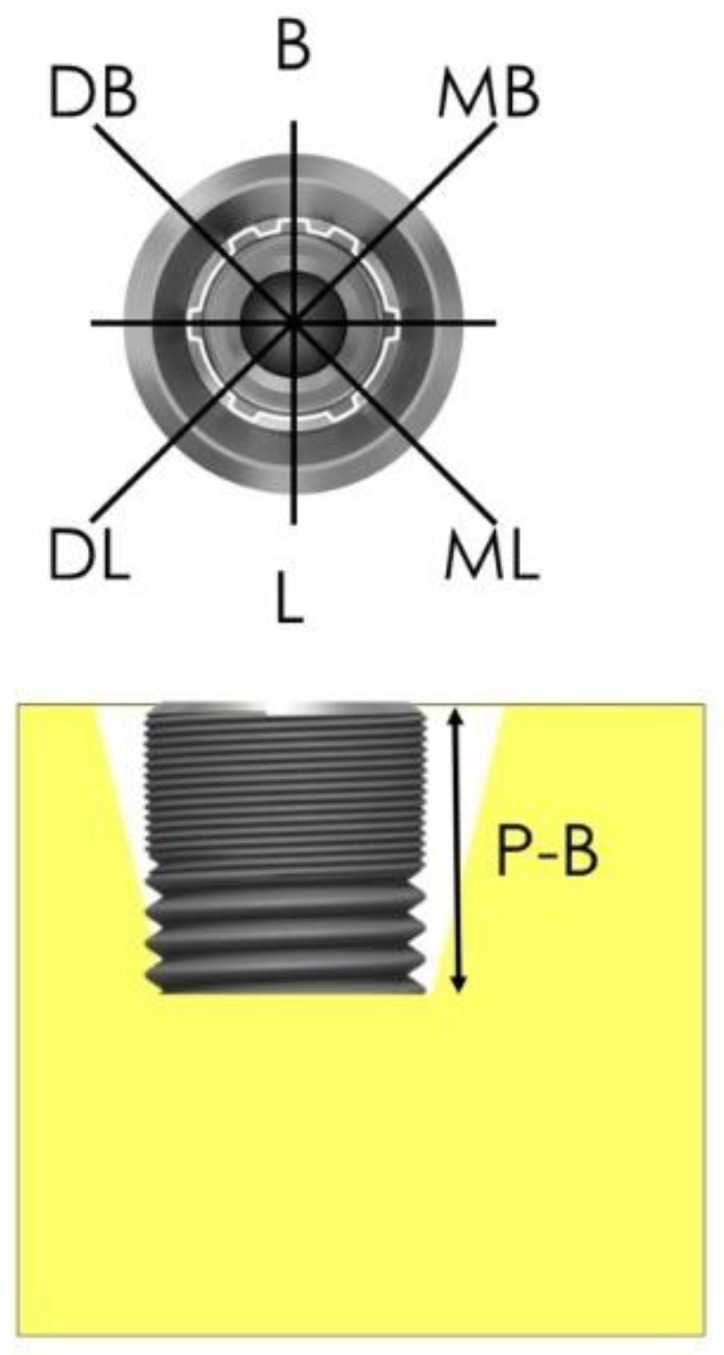
Bone defects were assessed at six defined points from implant shoulder to the most coronal position of the bone.

**Figure 4 jcm-08-01909-f004:**
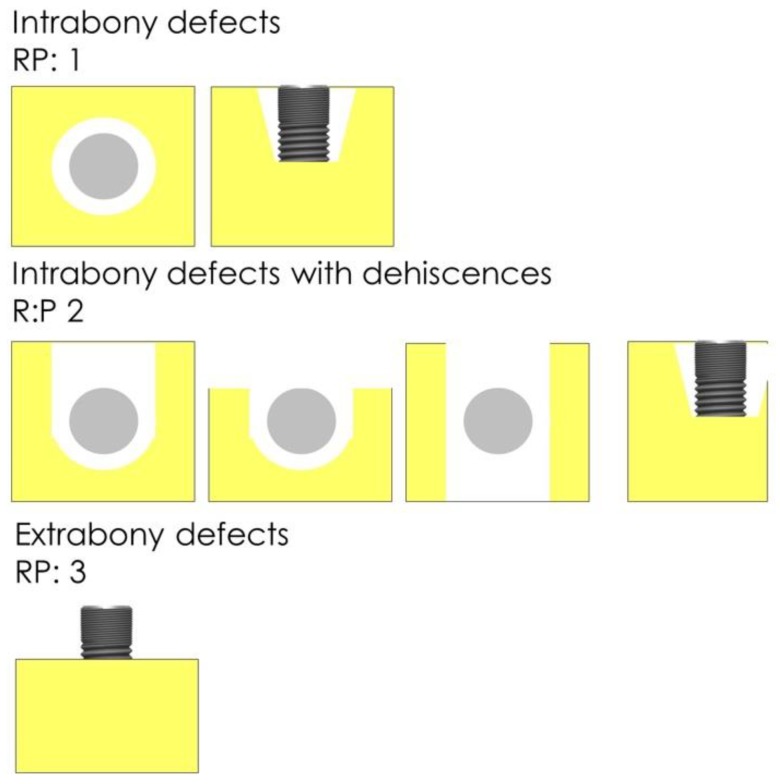
RP Classification of peri-implant bone defects based on risk-chance ratio of treatment.

**Figure 5 jcm-08-01909-f005:**
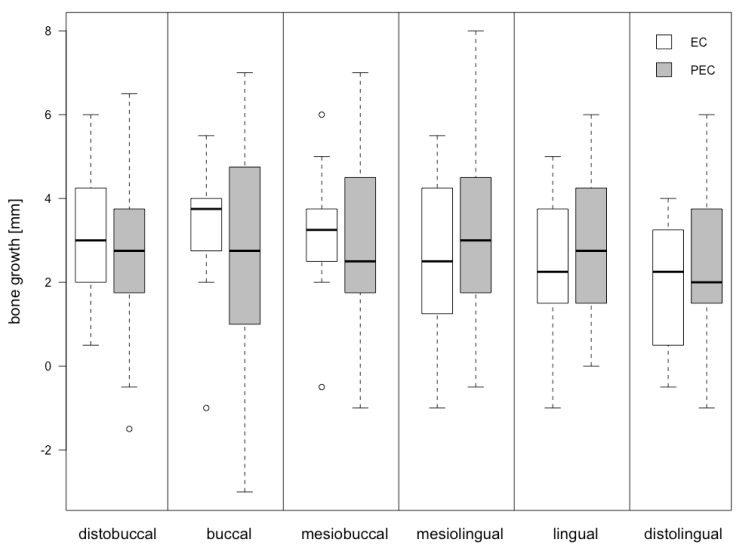
Boxplot indicating the distribution of the assessment points in EC and PEC.

**Figure 6 jcm-08-01909-f006:**
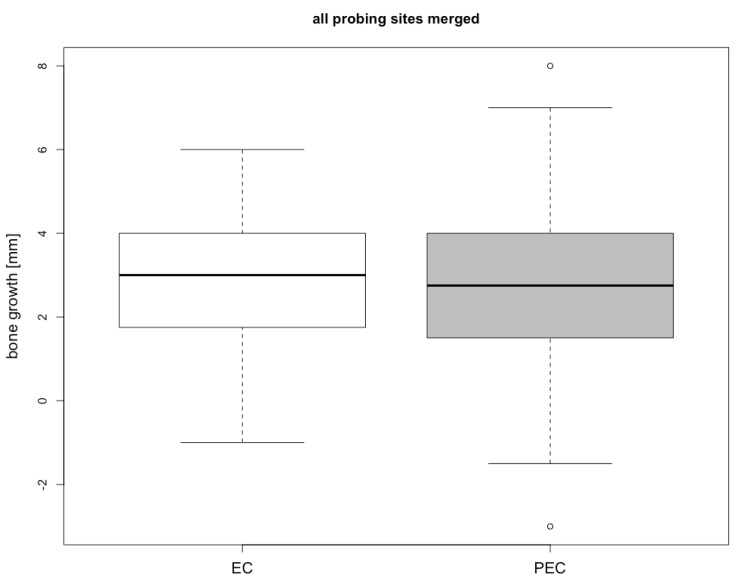
Boxplot indicating the merged distribution of the assessment points in EC and PEC.

**Figure 7 jcm-08-01909-f007:**
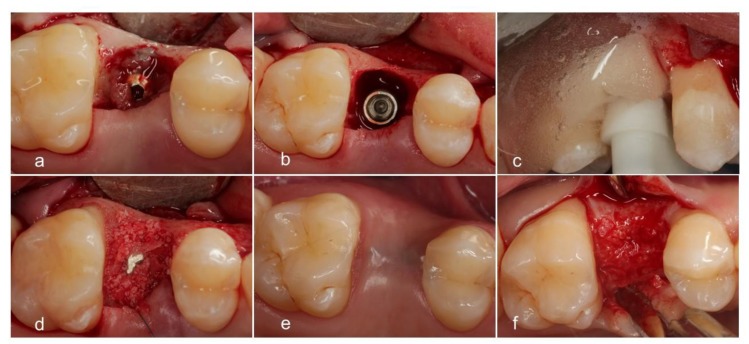
(**a**). A raised flap displays granulation tissue. (**b**). Deep peri-implant RP1 bone defect. (**c**). The spray-head is in place. The film of electrolyte is guided by a sponge. Hydrogen bubbles appear as a result of the process. (**d**). Defect augmented by a mixture of autogenous and natural bone mineral. (**e**). Healed defect after six months. (**f**). Solid bone overgrew the implant.

**Figure 8 jcm-08-01909-f008:**
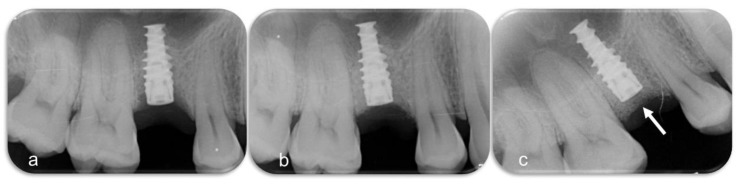
(**a**). The defect at baseline looks much less severe than in clinical reality. (**b**). Augmented defect. (**c**). Six months of healing.

**Table 1 jcm-08-01909-t001:** Qualitative baseline data indicating homogenous data.

General Information	Specific Information	*n*/Mean Years	Percentage
gender	female	12	50.00%
	male	12	50.00%
age	female	59.2 y	
	male	51.4 y	
jaw	maxilla EL	4	16.67%
	maxilla PEL	8	33.33%
	mandible EL	8	33.33%
	mandible PEL	4	16.67%
	maxilla total	12	50.00%
	mandible total	12	50.00%
smokers	EL	3	12.50%
	PEL	1	4.17%
BoP		24	100.00%
pus		24	100.00%

**Table 2 jcm-08-01909-t002:** Bone gain and differences between EC and AFL group.

Location	EC Group [mm]	PEC Group [mm]	Differences	
			AEL-EL	*p*-Value
db	3.00 ± 1.67	2.50 ± 2.10	−0.50	0.52
b	3.25 ± 1.63	2.83 ± 2.94	−0.42	0.71
mb	3.17 ± 1.61	3.00 ± 2.12	−0.17	0.83
ml	2.58 ± 1.84	3.13 ± 2.25	0.55	0.53
l	2.29 ± 1.79	2.96 ± 1.86	0.67	0.38
dl	1.96 ± 1.57	2.46 ± 1.83	0.50	0.48
Median	2.71 ± 1.70	2.81 ± 2.15	0.10	0.87

**Table 3 jcm-08-01909-t003:** Allocation of different implant types.

Type of Implant	Number
Astra TX	5
Astra EV	2
Straumann tissue level	2
Straumann bone level	1
Conelog	2
Camlog	2
Ankylos	2
Sky	1
Branemark	2
Xive	1
Steri Oss	1
Zimmer	2
Nobel Active	1

**Table 4 jcm-08-01909-t004:** Bone gain in relation to RP class.

RP Class	Bone Gain
RP1	4.02 ± 0.96
RP2	2.64 ± 1.58
RP3	2.34 ± 1.58

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
