# Peer review of "Treatment of Peri-Implantitis—Electrolytic Cleaning Versus Mechanical and Electrolytic Cleaning—A Randomized Controlled Clinical Trial—Six-Month Results"

_jcm, 2019, doi:10.3390/jcm8111909_

Round 1

Reviewer 1 Report

After reading the revised article and the authors replies to reviewer comments and suggestions, I recommend publication of this article as is. The manuscript and the understanding of the methodology used is now substancially improved.

Reviewer 2 Report

The manuscript is improved and may be accepted. In futire studies Control Groups are strongly recommended.

This manuscript is a resubmission of an earlier submission. The following is a list of the peer review reports and author responses from that submission.

Round 1

Reviewer 1 Report

I find unclear the method of assessing bone level changes, please explain it in more detail (procedures and measurements in Materials and Methods).

Reviewer 2 Report

This manuscript deals with a randomized clinical trial assessing the 6-month outcomes following therapy of periimplantitis lesions using either an electrolytic method (EC) or a combination of powder spray and electrolytic method (PEC). The results show no statistically significant difference between EC and PEC methods. Significant clinical bone fill was observed for all implants and complete regeneration of bone was achieved in half of the implants. The conclusion was that electrolytic method needs no further mechanical cleaning by powder spray, and a complete re-osseointegration in periimplantitis is possible.

Recommendation: Accept publication after a minor revision.

Comments: This manuscript is well authored, concise in format and sufficienly referenced.  The statistical evaluation is very good but the conclusions uncertain. This referee could not relate the present results to what can be achieved by the use other PI cleaning methods. Thus, the experimental planning suffers from the lack of comparison with another method than the EC method alone or in combination with powder spray. This makes it difficult to evaluate the significance of the results. Also, the results using the EC method only vs. the combination of EC+powder spray (PEC) do not differ. Which is the best? Or are the methods equally efficient? With this said, this manuscript can be accepted for publication upon a minor revision, se below commments.

Line 7. The word "and*"? Is the another author missing? Please correct.

Figure 1. The graph is difficult to understand. Please, improve it e.g. by clearly indicating what is what in text.

Discussion section. Discuss briefly the present results in comparisons to results by others and using other PI cleaning methods (mechanical, UV, etc). This would improve this article.

Reviewer 3 Report

The paper evaluates a timely and interesting topic. However, there are some sever shortcomings with the study design;

No control is included thus the clinical efficacy of the electrolytic cleaning is not evaluated only no effect was found for additional powder cleaning. A control group without EC is particularly important since other authors find mechanical brushing and saline being the most efficient method to decontaminate an implant surface. The follow-up period is only 6 months. As the authors themselves write, a 100% recurrence of periimplantis have been reported after 1 year, thus the minimum evaluation time ought to be 1 year.

Further comments; In the abstract it is written 1 implant was removed due to pus formation which seems to be a very odd reason for removing an implant. Later in the text more information is found, this implant was obviously mal positioned- maybe that was the main or even the only reason for the failure?

In the abstract it is also written “ a significant clinical bone fil was observed” How can bone fill ever be evaluated clinically? Maybe just a thin shell of bone and no fill?

In the introduction the authors blame unexperienced dentist to infect the implants that will eventually lead to a failure. With the knowledge we have today this statement seems a bit ignorant, the literature provides with evidence for a multifactorial explanation, infections be just one by many others. The authors seem to assume marginal bone loss eventually always leads to implant loss, this is according to existing literature not correct.

A pre-treatment using powder spray and chlorhexidine was used, why this? The risk for introducing confounding factors is high.

Re-osseointegration is claimed but cannot be evaluated clinically or with X-rays. A change in definition does not help.

A new classification included in the manuscript is out of the focus for the paper and not evaluated.

Overall the standard deviations are high, indicative of a too small sample size.